# Neurotrophic Effects of *Foeniculum vulgare* Ethanol Extracts on Hippocampal Neurons: Role of Anethole in Neurite Outgrowth and Synaptic Development

**DOI:** 10.3390/ijms252312701

**Published:** 2024-11-26

**Authors:** Sarmin Ummey Habiba, Ho Jin Choi, Yeasmin Akter Munni, In-Jun Yang, Md. Nazmul Haque, Il Soo Moon

**Affiliations:** 1Department of Anatomy, College of Medicine, Dongguk University, Gyeongju 38066, Republic of Korea; sarmin.ummey.habiba07@gmail.com (S.U.H.); chjack@naver.com (H.J.C.); yeasminakteracce@gmail.com (Y.A.M.); 2Medical Institute of Dongguk University, Gyeongju 38066, Republic of Korea; 3Department of Physiology, College of Korean Medicine, Dongguk University, Gyeongju 38066, Republic of Korea; injuny@dongguk.ac.kr; 4Department of Fisheries Biology and Genetics, Patuakhali Science and Technology University, Dumki 8602, Bangladesh; habib.uni.ac.bd@gmail.com

**Keywords:** *Foeniculum vulgare* Mill (FVSE), neurotrophic effects, neurite outgrowth, anethole, hippocampal neurons, synaptic development, neurotrophin signaling pathway, PI3K-AKT signaling pathway, neurodegenerative diseases

## Abstract

*Foeniculum vulgare* Mill, commonly known as fennel, is an aromatic herb traditionally used for culinary and medicinal purposes, with potential therapeutic effects on neurological disorders. However, limited research has focused on its neurotrophic impact, particularly on neuronal maturation and synaptic development. This study investigates the neurotrophic effects of *F. vulgare* ethanol extracts (FVSE) on the maturation of rat primary hippocampal neurons. Results show that FVSE and its prominent component, anethole, significantly promote neurite outgrowth in a dose-dependent manner. Optimal axonal and dendritic growth occurred at concentrations of 40 µg/mL FVSE and 20 µM anethole, respectively, without causing cytotoxicity, underscoring the safety of FVSE for neuronal health. Additionally, FVSE enhances the formation of synapses, essential for neuronal communication. Network pharmacology analysis revealed that FVSE components influence critical neurotrophic pathways, including PI3K-AKT and Alzheimer’s disease pathways. Specifically, FVSE modulates key proteins, including tropomyosin receptor kinase (Trk), glycogen synthase kinase 3 (GSK3β^ser9^), phosphatidylinositol 3-kinase (PI3K), and extracellular signal-regulated protein kinase (Erk1/2). Anethole was found to play a key role in regulating these pathways, which was confirmed by immunocytochemistry experiments demonstrating its effect on promoting neuronal growth and synaptic development. In conclusion, this study highlights the neurotrophic properties of FVSE, with anethole emerging as a critical bioactive compound. These findings provide valuable insights into the therapeutic potential of fennel in treating neurological disorders, offering a basis for future research into interventions promoting neuronal growth and survival.

## 1. Introduction

Neurodegenerative diseases (NDs), such as Alzheimer’s disease (AD) and Parkinson’s disease (PD), affect millions of individuals globally, reducing quality of life and placing substantial burdens on healthcare systems and caregivers. As the aging population grows, the incidence of NDs is expected to increase, emphasizing the need for effective prevention, early detection, and innovative treatments. A deeper understanding of the genetic, environmental, and lifestyle factors that contribute to NDs is essential for advancing therapeutic strategies to manage these conditions [1]. One of the early signs of neuronal dysfunction in NDs is the progressive loss of axons and dendrites, which leads to a reduction in synaptic connectivity and, ultimately, neuronal death. This breakdown of synaptic connections is closely associated with cognitive decline and memory deficits, highlighting the importance of developing therapies aimed at preserving synaptic integrity and promoting neuronal growth [2]. Moreover, chronic neuroinflammation driven by prolonged activation of glial cells exacerbates neuronal damage, further underscoring the need for neuroprotective strategies that can also mitigate inflammation alongside neurotrophic therapies [3,4].

A decline in neurotrophic factors (NTFs), including nerve growth factor (NGF) and brain-derived neurotrophic factor (BDNF), is a hallmark of NDs. These NTFs are essential for neuronal survival, synaptic plasticity, and cognitive function across various neurodegenerative conditions, not limited to Alzheimer’s disease (AD) [5]. In Parkinson’s disease, reduced levels of BDNF have been linked to dopaminergic neuronal loss, underscoring its broader impact on neuronal health [6]. Similarly, Huntington’s disease exhibits dysregulation of BDNF, contributing to the deterioration of striatal neurons [7]. Restoring or enhancing neurotrophic factor (NTF) signaling has the potential to slow disease progression by promoting neuronal growth, synaptic repair, and survival [8]. Thus, targeting NTFs or mimetics that can activate neurotrophic pathways holds promise for treatments aimed at improving cognitive function and quality of life in ND patients [9]. Natural phytochemicals have garnered attention for their potential to support neuronal health with lower toxicity profiles than many synthetic drugs. Phytochemicals can interact with neuroinflammatory mediators [10] and neurotrophic factors (NGF, BDNF, NT-3, and NT-5) [11,12], making them promising candidates for therapeutic applications. Specifically, some phytochemicals are known to activate key neuroprotective pathways, such as the PI3K-AKT and Erk1/2 pathways, which are essential for neuronal survival, differentiation, and synaptic plasticity. These pathways have been implicated in various NDs, making them attractive therapeutic targets [13,14,15].

*F. vulgare* Mill, commonly known as fennel, is an aromatic herb with a long history of use in culinary and medicinal practices, particularly in Ayurvedic and traditional Korean medicine [16,17,18,19]. Fennel possesses a wide range of biological activities, including antioxidant, anti-inflammatory, antitumor, and estrogenic effects, which contribute to its health benefits [20]. Recent studies have demonstrated that *F. vulgare* may help alleviate memory deficits and reduce stress, potentially due to its neuroprotective and anti-stress properties [21]. For instance, fennel has been shown to reverse memory impairment induced by scopolamine in rats, suggesting its potential as a natural cognitive enhancer [22]. In addition, *F. vulgare*’s antioxidant and anti-inflammatory properties are believed to support neuroprotection by modulating oxidative stress and inflammation, both of which play key roles in the pathogenesis of NDs.

Identifying the molecular mechanisms by which fennel exerts its neuroprotective effects could pave the way for developing novel plant-based interventions for NDs [8]. Recent research has identified several bioactive compounds within *F. vulgare* ethanol extracts (FVSE), including anethole, 2-propanone, 1-(4-methoxyphenyl)-, and para-anisaldehyde diethyl acetal [23], which may influence neurotrophic pathways such as PI3K-AKT and Erk1/2 [1,24]. Some of these compounds are recognized for their anti-oxidative effects, while others are believed to have direct neurotrophic effects, providing a multifaceted approach to neuroprotection [23,25,26,27,28,29,30].

This study aims to investigate the neurotrophic effects of FVSE on primary hippocampal neurons, focusing on its role in promoting neuronal differentiation, neuritogenesis, and synaptogenesis. Specifically, we hypothesize that FVSE enhances neurotrophic signaling pathways, thereby supporting neuronal growth and synaptic connectivity while protecting against neurodegenerative mechanisms. Using a combination of in vitro, in vivo assays, and in silico analyses, we explore how FVSE modulates neurotrophic signaling pathways to enhance neuronal development and connectivity. These findings may provide valuable insights into the neuro-pharmacological potential of FVSE, setting the stage for future natural therapeutic interventions against NDs.

## 2. Results

### 2.1. Determination of the Optimal Concentration of FVSE for Neurite Outgrowth Stimulation

The effects of FVSE on hippocampal neurons derived from embryonic day 19 (E19) rats were investigated by treating neurons with varying concentrations of FVSE. To identify the optimal concentration for promoting neurite outgrowth, neurons were cultured and exposed to different FVSE concentrations for three days in vitro (DIV3). Morphometric parameters, including the number and length of primary neurites, were evaluated (Figure 1). The dose-dependent effects of FVSE on neuronal growth are depicted in Figure 2A through bright field imaging (Appendix A). Statistical analysis revealed that FVSE at 40 μg/mL significantly enhanced all growth parameters compared to other concentrations and the control group. Specifically, treatment with 40 μg/mL FVSE resulted in a 1.5-fold increase in the number of primary neurites compared to the control (Figure 2B-a). Importantly, both the primary neurite length and the length of the longest neurite were approximately two-fold greater at this concentration compared to the vehicle control (Figure 2B-b,c). Based on these findings, 40 μg/mL was determined to be the optimal concentration for FVSE, and this dose was used for subsequent experimental investigations. These results suggest that FVSE significantly promotes neurite outgrowth in a dose-dependent manner, with 40 μg/mL providing the most effective stimulation.

### 2.2. FVSE Impact on Neuronal Viability

Additional analyses were performed to evaluate the effect of FVSE on the viability of hippocampal neurons derived from E19 rats using the trypan blue staining assay, a commonly used method for assessing cell viability. During the first eight days of incubation, cells treated with FVSE exhibited enhanced viability. Notably, FVSE at concentrations ranging from 10 to 60 µg/mL demonstrated neuroprotective effects by preventing cell death in a dose-dependent manner. At the highest concentration tested, the rate of cell viability was 96% (Figure 3). These findings indicate that FVSE is relatively safe for neurons at the tested doses.

### 2.3. Effects of FVSE on Neuronal Growth During Early Stages

Neuronal polarization is a critical step in neuronal differentiation, beginning with the formation of lamellipodia and progressing through distinct stages: no processes (Stage 1), minor processes (Stage 2), and axon formation (Stage 3). This study investigated the effects of FVSE extract on early neuronal differentiation by examining dissociated hippocampal neurons at DIV1 (24 h) and DIV2 (48 h) following FVSE treatment. To assess developmental stages, immunostaining was performed using anti-MAP2 (green; a somatodendritic marker) and anti-Tau (red; an axonal marker) antibodies, allowing for a comparison between vehicle-treated and FVSE-treated groups. Our analysis revealed significant differences in neuronal polarization between FVSE-treated and control-treated neurons (Figure 4A-a,B-a), indicating that FVSE promotes early neuronal maturation. Specifically, FVSE treatment resulted in increased axonal sprouting at Stage 3, with approximately 40% of neurons showing sprouting within 24 h in the FVSE-treated culture, compared to only 10% in the control group (Figure 4A-b). After 48 h, approximately 90% of FVSE-treated neurons reached Stage 3, demonstrating extensive axonal sprouting, while only about 15% of neurons in the control group reached this stage (Figure 4B-b). These findings underscore the ability of FVSE to promote neurite outgrowth and accelerate early neuronal differentiation.

### 2.4. Impact of FVSE on Axonal Development

This study assessed the effects of FVSE on axonal growth and differentiation in hippocampal neurons over a five-day treatment period. Quantitative morphometric analyses demonstrated that FVSE treatment significantly enhanced axonal development, as confirmed by Ankyrin G immunostaining, which selectively marks axons and distinguishes them from dendrites (Figure 5A,B). Results showed a 56.25% increase in axonal length in FVSE-treated neurons compared to vehicle-treated controls (Figure 5C-a), suggesting robust axonal elongation facilitated by FVSE. Moreover, FVSE treatment significantly enhanced axonal branching complexity. The frequency of primary and secondary axonal branches increased by 87.5% in FVSE-treated neurons, indicating a more intricate axonal network compared to controls (Figure 5C-b). Correspondingly, the length of primary branches doubled, and secondary branches were approximately 2.5 times longer in FVSE-treated neurons than in controls (Figure 5C-c). While tertiary branch numbers increased notably in FVSE-treated neurons, their lengths remained comparable to those observed in control neurons (Figure 5C,D). Furthermore, Sholl analysis further confirmed the effects of FVSE on axonal complexity, showing a significant increase in the number of axonal intersections within FVSE-treated neurons (Figure 5A). The peak of axonal intersections and branch point densities was observed at 320 µm and 200 µm from the cell center in FVSE-treated neurons, respectively, compared to 290 µm and 170 µm in control neurons (Figure 5D-a,b). These findings highlight FVSEs role in enhancing axonal outgrowth and structural complexity, suggesting its potential utility in promoting neural network formation and connectivity essential for neuronal function.

### 2.5. Improvement of Dendritic Arborization by FVSE

Neuronal development progresses rapidly after polarization, especially with the growth of dendritic tree-like structures, which typically occur between days three and seven during Stage 4. Building on this, our research explored FVSEs ability to enhance dendritic arborization at an early developmental stage, specifically on DIV5, as shown in Figure 5A. Our results demonstrated that FVSE treatment significantly increased both the number and total length of primary dendrites (Figure 6A-a,b). The number of primary dendrites increased by approximately 50% compared to the control group, while the total dendritic length increased by about 90%. To add to this, both the number and total length of primary and secondary branches were significantly greater in the FVSE-treated group compared to controls (Figure 6A-c,d). To further assess dendritic complexity, we employed Sholl analysis, which quantifies dendritic branching by measuring intersections with concentric circles at specified distances from the soma. This analysis revealed that dendritic intersections in FVSE-treated neurons extended up to 110 μm from the soma, compared to 60 μm in control-treated neurons (Figure 6B-a). In addition, the furthest dendritic branching points in FVSE-treated neurons were recorded at 80 μm, extending 50 μm beyond those of the control group (Figure 6B-b). These findings highlight FVSEs significant role in promoting dendritic arborization, suggesting its potential to enhance neuronal interconnectivity during early developmental stages.

### 2.6. Impacts of FVSE on Neuronal Synapse Formation

This study evaluated the effects of FVSE on synapse formation by analyzing synaptic markers and receptor subunits critical for synaptic connectivity and plasticity. Neuronal cultures were maintained for 16 days and labeled with the presynaptic marker SV2 and the postsynaptic marker PSD-95, followed by epifluorescence microscopy analysis (Figure 7A-a–c). FVSE-treated neurons showed a significant increase in the density of SV2 and PSD-95 puncta compared to controls, as depicted in Figure 7B-a–c. To accurately assess synapse formation, presynaptic and postsynaptic puncta were overlaid to identify co-localization. FVSE-treated neurons demonstrated a notable increase in co-localized puncta density within a 50 μm segment, indicating enhanced synaptic connectivity (Figure 7B-a–c). On top of that, FVSE treatment significantly upregulated the NMDA receptor subunits GluN2A and GluN2B, which are essential for synaptic plasticity. Immunoblotting results (Figure 7C-a–c) revealed a substantial increase in GluN2A and GluN2B levels in FVSE-treated neurons compared to controls (** *p* < 0.01; Figure 7D-a–c). Fluorescence microscopy confirmed that these NMDA receptor subunits co-localized with SV2 at synaptic sites, resulting in an increased density of GluN2A-SV2 and GluN2B-SV2 puncta in FVSE-treated neurons (Figure 7A-a,b,B-a,b). This co-localization suggests that FVSE enhances the integration of GluN2A and GluN2B into synaptic structures, thereby contributing to synaptic stability and flexibility. Specifically, GluN2A is associated with synaptic stabilization and long-term potentiation, while GluN2B supports synaptic adaptability, indicating that FVSE promotes both the stability and adaptability of synaptic networks, which are essential for learning and memory. Furthermore, FVSE treatment led to a 1.56-fold increase in PSD-95 expression compared to the control group (Figure 7C-c,D-c), reinforcing FVSEs positive impact on synaptic assembly and plasticity. Collectively, these results indicate that FVSE significantly enhances synaptic protein expression and assembly, underscoring its therapeutic potential for improving neural connectivity. This enhancement of synaptic components may contribute to cognitive resilience and neuroplasticity, highlighting FVSEs promise in supporting brain health.

### 2.7. In-Vivo Immunoblotting Analysis of FVSE

To examine the effects of FVSE on synaptic plasticity and connectivity, we conducted immunoblotting analysis on hippocampal tissue lysates from control and FVSE-treated female mice after five weeks of treatment. This analysis focused on the expression of key synaptic markers and NMDA receptor subunits to provide insights into FVSEs potential neuroprotective effects. Representative Western blot images (Figure 8) reveal significant increases in the expression levels of GluN2A, GluN2B, and PSD95 in the FVSE-treated group compared to the control group, with α-tubulin used as the loading control. Quantitative analysis showed that GluN2A expression was approximately doubled in FVSE-treated mice relative to controls (Figure 8A,B-a). Similarly, GluN2B expression exhibited a marked and statistically significant increase (Figure 8A,B-b), as did PSD95, a key protein associated with synaptic connectivity (Figure 8A,B-c). The quantitative evaluation confirmed that these changes were statistically significant (mean ± SEM, *** *p* < 0.001) across all markers tested, indicating that FVSE robustly enhances the expression of synaptic proteins. The upregulation of GluN2A and GluN2B suggests that FVSE positively modulates NMDA receptor function, which plays a crucial role in synaptic plasticity and memory processes. Similarly, the increase in PSD95 expression indicates enhanced synaptic connectivity, likely due to improved clustering of synaptic receptors, which is essential for maintaining synaptic strength and plasticity—processes that are often compromised in neurodegenerative conditions.

### 2.8. Analysis of FVSE Bio-Active Compounds and Neuroprotective Targets Network

The analysis of FVSE focused on identifying its bioactive compounds and their potential neuroprotective targets. A Venn diagram (Figure 9A) revealed an overlap of 376 genes between those associated with the bioactive components of FVSE (617 genes) and those related to neuronal development (3787 genes). This significant overlap suggests that the bioactive compounds in FVSE may influence pathways essential for neuronal growth and differentiation. Further analysis using protein-protein interaction (PPI) network mapping (Figure 9B,C) identified key hub proteins integral to neuroprotective mechanisms, indicating that FVSE compounds may interact with these critical proteins to exert neuroprotective effects. A bar chart (Figure 9B) lists the top 20 genes associated with neuronal development, highlighting their importance in the context of FVSEs potential therapeutic benefits. Noteworthy genes, such as Bcl2, GSK3βser9, Trk-A, Trk-B, and BDNF, are emphasized for their roles in promoting neuronal survival and plasticity.

To further understand FVSEs biological impact, enrichment analyses were performed. The results, visualized through dot plots (Figure 9D-a–d), show that the common target genes are enriched across multiple biological processes, cellular components, molecular functions, and pathways. Specifically, these genes are involved in processes such as synaptic signaling, regulation of cell death, and response to various stimuli (Figure 9D-a). Enriched cellular components include the presynaptic membrane, dendritic tree, and neuronal cell body (Figure 9D-b). Molecular functions involve signaling receptor activity and transmembrane signaling receptor activity (Figure 9D-c).

KEGG pathway analysis (Figure 9D-d) revealed significant involvement in pathways related to neurotrophic signaling, PI3K-AKT signaling, and Alzheimer’s disease, among others. The enrichment of the neurotrophic signaling pathway is particularly notable, supporting the hypothesis that FVSE compounds may promote neuroprotection and enhance neuronal connectivity by modulating these critical pathways (Appendix A). These findings suggest that FVSE holds promise as a therapeutic agent for neurodegenerative diseases, potentially offering a multifaceted approach to supporting neuronal health and neuroprotection.

### 2.9. Effect of Anethole and FVSE on Neurite Growth

The effects of anethole and FVSE on neuronal process development were assessed and compared with those of several positive controls, including Scoparone (37.5 μM), Linalool (10 μM), and Stigma-sterol (75 μM). As shown in Figure 10A, neurons treated with Anethole (20 μM) (Appendix A) and FVSE (40 μg/mL) exhibited significantly more extensive neurite outgrowth compared to the control group. Quantitative analysis revealed that both anethole and FVSE significantly increased the number of primary neurites per neuron (Figure 10B-a), with FVSE treatment showing a comparable or greater effect than the positive controls. In parallel, the length of primary neurites was significantly longer in neurons treated with FVSE and anethole, with FVSE demonstrating the most pronounced effect (Figure 10B-b). The length of the longest neurite was also significantly greater in the FVSE-treated group compared to the control and positive control groups, underscoring FVSEs potent ability to promote neurite extension (Figure 10B-c). These findings suggest that both anethole and FVSE have strong neurotrophic effects, effectively promoting neurite outgrowth and elongation, surpassing the performance of the positive controls. These results highlight FVSEs potential as a promising natural agent for enhancing neuronal growth and development.

### 2.10. Immunofluorescence Analysis of Neurotrophin Signaling Proteins

To investigate the neuroprotective effects of FVSE ethanol extract and anethole, we conducted immunocytochemistry on primary cultured hippocampal neurons to evaluate the expression of Neurotrophin signaling proteins, including NGF, BDNF, Trk-A, Trk-B, and GSK3β^ser9^ (Figure 11). Representative images show neurons labeled with α-tubulin (green) as a cytoskeletal marker alongside specific neurotrophin markers (red), with merged images indicating regions of co-localization (Figure 11A). Quantitative analysis revealed a significant increase in the fluorescence intensity of NGF, BDNF, Trk-A, and Trk-B in neurons treated with FVSE and anethole compared to control neurons (Figure 11B, panels a–d). In contrast, GSK3β^ser9^ expression was significantly decreased in the FVSE- and anethole-treated groups, suggesting suppression of the GSK3β^ser9^ signaling pathway (Figure 11B, panel e). These findings indicate that FVSE ethanol extract and anethole enhance Neurotrophin signaling, thereby promoting neurite development in primary hippocampal neurons.

## 3. Discussion

*F. vulgare* has long been used in traditional medicine for various therapeutic purposes, including treating neuronal disorders. Recent studies support its neuroprotective effects, reinforcing its historical application for neuronal health [31,32]. However, its potential for neurotrophic effects, especially in promoting neuronal growth and maturation, has not been fully explored. In this study, we investigated the neurotrophic potential of FVSE using dissociated hippocampal cultures as an in vitro model. Our findings indicate that FVSE significantly enhances neuronal survival, axon-dendritic arborization, synaptogenesis, and modulates key signaling pathways associated with these morphological changes in cultured rat hippocampal neurons.

Our results demonstrated that FVSE treatment at 40 µg/mL significantly increased the number and length of primary neurites, especially during early neuronal differentiation stages [33,34]. This effect was especially notable during the critical stages of neuronal polarization, where FVSE facilitated the transition from lamellipodia formation (Stage 1) to the establishment of minor processes (Stage 2), and finally to axon elongation (Stage 3) [35,36]. These processes are vital for neuronal function and connectivity, and FVSEs capacity to accelerate these stages may address developmental delays commonly associated with neurodegenerative diseases (NDs) [37].

Biochemical validation of the enriched signaling pathways identified in the KEGG analysis, such as the Neurotrophin signaling pathway, PI3K-AKT signaling pathway, and Alzheimer’s disease pathway, strengthens our findings. For instance, FVSE treatment significantly increased the expression of neurotrophic factors NGF and BDNF and their receptors Trk-A and Trk-B [38], as demonstrated by immunocytochemical analysis (Figure 11A). These proteins are central to the Neurotrophin signaling pathway, highlighting FVSEs potential in enhancing neuronal survival and synaptic plasticity. Quantification of fluorescence intensity (Figure 11B) further supports these findings, showing significant upregulation of these key neurotrophic markers. Similarly, the upregulation of GSK3β^ser9^ and AKT1 indicates FVSEs ability to activate the PI3K-AKT pathway, which is critical for promoting cell survival and axonal growth, while inhibiting pro-apoptotic mechanisms [39]. These findings are consistent with the KEGG analysis, which identified AKT1, GSK3β^ser9^, and Bcl2 as pivotal genes regulated by FVSE.

The Alzheimer’s disease KEGG pathway enrichment highlights FVSEs ability to modulate critical processes involved in Alzheimer’s pathology, such as amyloid-beta clearance, mitochondrial function, and oxidative stress response [40]. Increased expression of NMDA receptor subunits GluN2A and GluN2B, along with synaptic markers PSD95 and SV2 (Figure 7 and Figure 8), indicates FVSEs potential to restore synaptic connectivity often compromised in Alzheimer’s disease [41]. The increased expression of HIF1A, a hypoxia-inducible factor, suggests that FVSE enhances neuronal resilience by reducing oxidative stress and preserving mitochondrial function, which is critical in Alzheimer’s disease pathology [40]. Additionally, the upregulation of TP53, a regulator of apoptosis and DNA repair, suggests that FVSE could prevent neuronal death by promoting cellular repair mechanisms under stress conditions [42]. Moreover, PTGS2 (COX-2), a known mediator of inflammation, was modulated by FVSE, highlighting its potential to reduce neuroinflammation and improve neuronal survival [43]. Together, these findings suggest that FVSE not only enhances synaptic plasticity but also modulates pathways implicated in neurodegeneration, thereby providing cognitive resilience and neuroprotection. These findings align with KEGG enrichment analysis, which identified AKT1, BDNF, TP53, GSK3β^ser9^, and HIF1A as pivotal regulators of the Neurotrophin, PI3K-AKT, and Alzheimer’s disease pathways. The combination of these biochemical findings and KEGG enrichment underscores FVSEs therapeutic potential in targeting Alzheimer’s disease through multiple mechanisms, and enhanced synaptic connectivity.

In addition to its direct effects on neurons, FVSE may also interact with glial cells, which play essential roles in the neuronal microenvironment. Astrocytes provide metabolic and structural support to neurons, facilitate synaptogenesis, and maintain neurotransmitter homeostasis, while microglia mediate synaptic pruning, immune defense, and resolution of neuroinflammation [44,45,46]. Oligodendrocytes contribute by promoting myelination, which is critical for efficient neural communication [47,48]. Although glial cells were not the primary focus of this study, the anti-inflammatory and neuroprotective properties of FVSE suggest that its bioactive compounds may positively influence glial and immune cell functions [49,50]. Future studies incorporating co-culture systems of neurons and glial cells could provide deeper insights into these broader interactions. Such studies could reveal whether FVSEs bioactive compounds enhance astrocytic neurotrophic support, regulate microglial inflammatory responses, or promote oligodendrocyte myelination, thereby expanding its therapeutic potential.

Additionally, FVSE treatment increased the expression of GluN2A and GluN2B subunits of NMDA receptors, which are crucial for synaptic plasticity, long-term potentiation, and cognitive processes like learning and memory [51,52,53]. GluN2A is associated with synaptic strengthening, while GluN2B is linked to synaptic flexibility, indicating that FVSE may enhance both stability and adaptability of synaptic connections. This suggests that FVSE could improve synaptic function and promote cognitive resilience in neurodegenerative conditions. Moreover, FVSE treatment significantly increased synaptic density, indicated by the elevated expression of SV2 and PSD95, which are vital for synaptic transmission and plasticity. SV2 is involved in neurotransmitter release, while PSD95 clusters neurotransmitter receptors, suggesting that FVSE enhances both pre- and postsynaptic structures, strengthening the synaptic connections often compromised in NDs.

GC-MS analysis identified key bioactive compounds in FVSE, including anethole, silane, and para-anisaldehyde diethyl acetal, which regulate signaling pathways related to neuroprotection, neurodevelopment, and neuroplasticity [16,17]. Anethole, in particular, is known for its 1:50 anti-inflammatory and antioxidant properties, which could contribute to FVSEs neuroprotective effects [54,55]. KEGG pathway analysis revealed that FVSE significantly upregulates genes involved in Neurotrophin signaling and pathways relevant to Alzheimer’s disease, such as PI3K-AKT signaling [56]. The modulation of pathways involving PI3K [56], Trk-A [29], Trk-B [57], Bcl2, GSK3βser9 [1], and Erk1/2 [5,58] underscores FVSEs potential role in promoting neuronal survival and function, especially in managing neurodegenerative conditions like Alzheimer’s disease [59,60]. By modulating these pathways, FVSE may promote neuronal survival, synaptic plasticity, and neurogenesis, offering therapeutic potential in managing Alzheimer’s disease [5,7,55,61].

Key kinases such as Trk-B, GSK3βser9, and Erk1/2 play crucial roles in neuronal survival, synaptic plasticity, and cognitive function by influencing processes such as dendritic branching, axonal growth, and synaptic strength [61,62]. Upregulation of Erk1/2 and Trk-B suggests that FVSE could enhance neuronal resilience and cognitive function. GSK3βser9 is an important regulator of apoptosis and neurogenesis [63,64,65,66], and its modulation by FVSE could prevent neuronal death, preserving cognitive functions, especially in NDs where GSK3βser9 dysregulation is common [67]. The upregulation of BDNF and AKT1 in the PI3K-AKT pathway further supports FVSEs role in promoting neurogenesis and synaptic plasticity [5]. By enhancing CREB1 activity, FVSE may improve memory formation and cognitive functions [65]. Likewise, FVSE modulates TP53 and PTGS2, which are associated with stress responses and neuro-inflammation, providing neuroprotective effects. The increased expression of HIF1A suggests that FVSE may enhance resilience under metabolic stress, a common feature in neurodegenerative diseases [68,69,70].

Overall, FVSE promotes neuronal growth, survival, and synaptic function while providing protection against neurodegeneration. Protein-protein interaction (PPI) network analysis highlighted key genes, including AKT1, TP53, Trk-B, and BDNF, which are essential for neuronal survival and plasticity [71,72]. BDNF, a well-known neurotrophic factor, supports synaptic plasticity and cognitive functions [73,74]. (Findings are summarized and presented in Figure 12). The upregulation of these genes in response to FVSE treatment underscores its potential as a natural therapeutic agent for neurodegenerative diseases (NDs) [75,76]. FVSE enhances neuronal growth, synaptic connectivity, and viability, making it a promising candidate for maintaining cognitive function in neurodegeneration. The protective effects of FVSE on neuronal survival are likely due to the antioxidant and anti-inflammatory properties of its bioactive compounds [77,78]. Building on these findings, FVSE appears to modify larger neuronal networks by enhancing synaptic connectivity and promoting neurogenesis [79,80]. Combined evidence from in vitro, in vivo, and in silico analyses suggests that FVSE has significant potential as a natural therapeutic agent for enhancing neuronal health and resilience, particularly in Alzheimer’s disease and other neurodegenerative conditions [1].

Future studies should incorporate validation of key proteins (AKT1, BDNF, TP53, and Trk-B) to confirm these observations at the biochemical level. Also, in vivo studies are essential to assess FVSEs safety, efficacy, and mechanisms of action in more complex biological systems, further establishing its therapeutic relevance in neurodegenerative diseases.

## 4. Methods and Materials

### 4.1. Preparation of F. vulgare Mill Ethanol Extracts (FVSE)

*F. vulgare* seeds (FVSE) were sourced from a farm in Gyeongsang Province, Republic of Korea. The seeds were carefully selected by an expert to ensure their suitability for use in hippocampal culture experiments. The following chemicals and reagents were used: trypsin inhibitor (Welgene, Gyeongsan, Gyeongsangbuk, Republic of Korea), Poly-D-lysine (Sigma-Aldrich, St. Louis, MO, USA), boric acid (Sigma-Aldrich, St. Louis, MO, USA, CAS 10043-35-3), Hank’s balanced salt solution (HBSS, Welgene, Gyeongsan, Gyeongsangbuk, Republic of Korea), dimethyl sulfoxide (DMSO, Sigma-Aldrich, St. Louis, MO, USA, CAS 67-68-5), and β-mercaptoethanol (Sigma-Aldrich, St. Louis, MO, USA, CAS 60-24-2). Additional components included Glutamax I (0.5 mmol/L, Gibco, Carlsbad, CA, USA), Penicillin-Streptomycin (Welgene, Gyeongsan, Gyeongsangbuk, Republic of Korea), B27 supplement (Gibco, Carlsbad, CA, USA), HEPES (Gibco, Carlsbad, CA, USA), Sodium Pyruvate (Dae-Jung, Republic of Korea; CAS 113-24-6), Neurobasal Medium (Gibco, Carlsbad, CA, USA), and Fetal Bovine Serum (Avantor®, Radnor, PA, USA).

The FVSE were air-dried at room temperature, ground into a fine powder using a grinder (HMF-340, Hanil Co., Seoul, Republic of Korea), and stored in airtight bags at −20 °C. This process was conducted to maintain the quality and consistency of the seeds for extraction. The seed powder was washed with distilled water and dried at room temperature before ethanol extraction. The dried seeds were then ground using liquid nitrogen and further processed. We used an extraction ratio of 1:50 with 95% ethanol, based on preliminary studies that indicated optimal extraction efficiency at this ratio for neuroprotective studies. The prepared seeds were mixed with the solvent, and the mixture was agitated overnight on an orbital shaker (VS-202D, Vision Scientific Co., Ltd., Seoul, Republic of Korea) at 200 RPM, at room temperature, and in the dark, to protect the integrity of the extract. After 24 h, the mixture was filtered through a sterile cotton filter to remove solid particles, and the supernatant was collected. The supernatant was concentrated under a vacuum using a nitrogen gas stream. The concentrated extract was then dissolved in DMSO at a final concentration of 8 mg/mL, a concentration chosen based on prior dose-response findings that indicated its effectiveness for in vitro studies on neuronal cells. The solution was stored in foil-wrapped vials at −20 °C to maintain stability until further use.

### 4.2. Primary Hippocampal Neuronal Culture and Treatment of FVSE

An in vitro study was conducted to evaluate the effects of FVSE on neurodevelopment in primary neuronal cells. All experimental procedures followed the guidelines of the Animal Care and Use Committee of Dongguk University College of Medicine (approval certificates IACUC-2023-06 and IACUC-2023-07), adhering to the Principles of Laboratory Animal Care (NIH, Washington, DC, USA). Pregnant E-17 Sprague-Dawley rats were housed under a 12-h light/dark cycle with continuous access to food and water for two days. On gestation day 19, primary hippocampal neurons were isolated from the fetuses following established protocols [28,29]. The pregnant rats were anesthetized using isoflurane, and embryos were collected post-sacrifice. Hippocampi were dissected from the embryos in cold Hank’s balanced salt solution (HBSS). After removing the meninges, the hippocampi were digested with 0.25% trypsin-EDTA at 37 °C for 12 min to isolate neurons. Cells were then dissociated by gentle trituration with fire-polished Pasteur pipettes. The dissociated cells were seeded onto Poly-D-lysine (PDL)-coated glass coverslips in 24-well and 6-well plates. For morphometric analysis, cells were plated at a density of 1.0 × 10^4^ cells/cm², while for synaptogenesis and viability assays in 24-well plates, the density was 2.0 × 10^4^ cells/cm². For Western blotting in 6-well plates, the cell density was 3 × 10^6^ cells/cm². B27-supplemented serum-free neurobasal medium was used as the plating medium, which was combined with FVSE extract, control, or other compounds before seeding the cells. Throughout the experimental period, fresh media was provided to the neurons every three days, and FVSE was administered at the experimental dose during each media change to evaluate its effects on neuronal development.

### 4.3. In Vivo Mice Experiment of FVSE

Institute of Cancer Research (ICR) female mice (5 weeks old) were divided into two groups (*n* = 4 in each group). Animal handling followed the guidelines of the Institutional Animal Care and Use Committee of Dongguk University (IACUC-2021-13). A suspension of FVSE and a control solution [1% (*v*/*v*) Tween 80 (Sigma-Aldrich, St. Louis, MO, USA), 1% (*v*/*v*) DMSO, and 0.9% (*w*/*v*) NaCl solution] were prepared as described previously [17]. Solutions were prepared daily and administered orally from the mating phase through the post-natal phase. The control group received the solution containing 1% (*v*/*v*) Tween 80, 1% (*v*/*v*) DMSO, and 0.9% normal saline, while the FVSE-treated group received 200 and 400 mg/kg of FVSE extracts orally (3 months) per day.

### 4.4. Neuronal Viability

On day eight of the in vitro experiment, neuronal viability was assessed using a trypan blue exclusion assay. Both control (DMSO-treated) and FVSE-treated neurons were stained with a 0.4% trypan blue solution. Neurons were incubated at 37 °C for 10–15 min, then washed with Dulbecco’s phosphate-buffered saline (D-PBS, Invitrogen). Under a bright-field microscope, non-viable neurons appeared dark blue as they absorbed the dye due to compromised plasma membranes, while viable neurons excluded the dye, remaining unstained. To determine cell viability, the percentage of live (unstained) neurons was calculated by counting 1000 total cells (stained and unstained) to ensure accuracy.

### 4.5. Immunocytochemistry

Immunocytochemistry was performed on neuronal cultures from DIV1 to DIV14 to assess neuronal morphology and synaptic architecture. Neurons grown on Poly-D-lysine-coated coverslips were washed twice with 1 × D-PBS to remove any residual media and fixed using 4% paraformaldehyde for 15 min to preserve cellular structures, followed by methanol treatment to enhance structural stability. Permeabilization with 0.1% Triton X-100 in PBS was performed to allow antibody access to intracellular components. To reduce non-specific binding, coverslips were incubated with a blocking solution containing 0.4% fish skin gelatin, 2.5% goat serum, and 0.2% Tween-20 in PBS for 1 h at room temperature. Primary antibodies, diluted in PBS with 1% bovine serum albumin (BSA) and 0.3% Triton X-100 to enhance antibody stability, were applied to the coverslips and incubated overnight at 4 °C for optimal binding. These antibodies targeted critical neuronal and synaptic markers: Anti-α-Tubulin subunit (1:100, Developmental Studies Hybridoma Bank, Iowa City, IA, USA) for cytoskeletal integrity, Rabbit anti-ankyrin G (1:50, Santa Cruz Biotechnology, CA, USA) for axon initial segment identification, Mouse anti-MAP2 (1:250, Sigma, St. Louis, MO, USA) to label dendritic structures, Rabbit anti-Tau (1:100, Ab-Frontier, Seoul, Republic of Korea) for axonal characterization, Mouse anti-SV2 (1:50, Developmental Studies Hybridoma Bank, Iowa City, IA, USA) for presynaptic terminals, Chicken anti-PSD95 (1:1000, provided by Dr. R.S. Walikonis, University of Connecticut, CT, USA) for postsynaptic densities, and Rabbit anti-GluN2A and GluN2B (1:500) to analyze NMDA receptor subunits involved in synaptic plasticity. Similarly, antibodies specific to neurotrophic signaling and synaptic function were included: BDNF (rabbit polyclonal, 1:100; Invitrogen, Carlsbad, CA, USA), NGF (1:500; Santa Cruz Biotechnology, Santa Cruz, CA, USA), Trk-A (rabbit polyclonal, 1:500; ABclonal, Wuhan, China), Trk-B (rabbit polyclonal, 1:500; ABclonal, Wuhan, China), and GSK3β^ser9^ (1:500; Cell Signaling Technology, Danvers, MA, USA). After washing to remove unbound primary antibodies, secondary antibodies conjugated with Alexa Fluor dyes were applied for 2 h and 30 min at room temperature to allow multi-color visualization of the targets. Specifically, Alexa Fluor 488-conjugated goat anti-mouse IgG, Alexa Fluor 568-conjugated goat anti-rabbit IgG, and Alexa Fluor 568-conjugated goat anti-chicken IgG (all at 1:500, Molecular Probes, Eugene, OR, USA) were selected based on compatibility with primary antibodies and chosen for their brightness and photostability to ensure high-quality imaging. Coverslips were then mounted on slides using anti-fade medium to preserve fluorescence. This immunocytochemistry protocol enabled detailed visualization of neuronal morphology and synaptic architecture, providing insights into cellular organization and functional responses to FVSE treatment, relevant to understanding neuronal development and synaptic function.

### 4.6. Western Blotting 

Neuronal cells were harvested from six-well cell culture plates and washed with cold 1× D-PBS. After rinsing, the cells were lysed in a protein extraction buffer containing 50 mM Tris-HCl (pH 8.0), 0.5% NP-40, 150 mM NaCl, 1% sodium deoxycholate, and 1% sodium dodecyl sulfate (SDS). A protease inhibitor cocktail was added to the lysis buffer, and the cell lysates were incubated on ice for 20 min. Following incubation, the lysates were centrifuged at 13,000 rpm for 15 min at 4 °C, and the resulting supernatant-containing proteins were collected. Protein concentration was determined using the Advanced Protein Assay Reagent, with bovine serum albumin (BSA) as the standard. Each cell lysate sample, containing 40 µg of protein, was mixed with 5× sample buffer (60 mM Tris-HCl, pH 6.8, 2% SDS, 14.4 mM β-mercaptoethanol, 25% glycerol, and 0.1% bromophenol blue) and boiled for 5 min to fully denature the proteins. The samples were then loaded onto a sodium dodecyl sulfate-polyacrylamide gel (SDS-PAGE) and separated by electrophoresis. The proteins were subsequently transferred to a Polyvinylidene difluoride (PVDF) membrane. To prevent non-specific binding, the PVDF membranes were blocked with 5% skim milk in Tris-buffered saline containing 0.05% Tween-20 (TBS-T) for one hour at room temperature. The membranes were then incubated overnight at 4 °C with primary antibodies specific to PSD95, GluN2A, GluN2B, and tubulin α-subunit, which served as a loading control. The primary antibodies used were chicken polyclonal UCT-C1 (1:1000 dilution) for PSD95 and rabbit polyclonal antibodies (1:1000 dilution) for GluN2A and GluN2B. For tubulin α-subunit, a mouse monoclonal antibody (12G10; 1:1000 dilution) was used as the loading control. Following incubation, the membranes were washed with Tris-buffered saline containing 0.05% Tween-20 (TTBS) and incubated with horseradish peroxidase-conjugated secondary antibodies (1:10,000 dilution) specific to rabbit and chicken IgG for 2 h at room temperature. Immuno-reactive bands were visualized using an Enhanced Chemiluminescence Detection Kit (West Save Gold; Ab-Frontier, Young Frontier, Seoul, Korea). The intensity of the protein bands was quantified using ImageJ software (ver. 1.49).

### 4.7. Imaging and Morphometric Analysis of Neuronal Morphology

Neuronal cells were imaged using a Leica DFC3000G fluorescence microscope (Leica Microsystems Ltd., Wetzlar, Germany) equipped with a Sony^®^ CCD monochrome sensor (1.3 megapixels). The imaging process was controlled by Leica Application Suite X (Las X Version: 3.7.2.22383) software, and images were further refined as needed using Adobe Photoshop 7.0 for clarity. For morphometric analysis, ImageJ (version 1.49) was used to quantify parameters such as the number of primary dendrites, total dendritic and axonal lengths, and branching patterns of both axons and dendrites. Neurite arborization analysis, including axonal and dendritic structures, was conducted using the Neurite Tracer Sholl plug-in (NIH, Bethesda, MD, USA), accessed on 28 December 2022. This analysis involved assessing intersections between neurites and concentric circles spaced 10 μm apart. Axonal and dendritic intersections were recorded at points where these structures crossed specific concentric circles and branching points were identified between adjacent circles. Notably, synaptic puncta images were captured using an Olympus BX53 microscope (Olympus, Tokyo, Japan) equipped with a DP74 1/1.2-inch color CMOS camera. These images were processed and analyzed using Cell Sens software (Version: 1.18).

### 4.8. A In Silico Network Pharmacology

#### 4.8.1. ADMET Analysis of GC-MS/HPLC Compounds

In a prior study, ethanol extracts of FVSE were analyzed using GC-MS and HPLC, identifying 40 compounds. The ADMET properties (Absorption, Distribution, Metabolism, Excretion, and Toxicity) of these compounds were evaluated to assess their drug-like characteristics, bioavailability, and safety profiles. Using in silico network pharmacology, the biological activities and potential molecular interactions of these compounds were explored to gain insights into their therapeutic potential. The 2D structures of the identified compounds were sourced from the PubChem database (https://pubchem.ncbi.nlm.nih.gov/, accessed on 16 August 2024), and their pharmacokinetic properties, including ADMET characteristics, were analyzed using the Swiss ADME online tool (http://www.swissadme.ch, accessed on 16 August 2024). These analyses helped predict bioavailability, toxicity, and potential suitability for therapeutic development [23].

#### 4.8.2. Screening of the Common Target Genes

Common target genes were predicted using various databases, including SwissTargetPrediction (http://www.swisstargetprediction.ch/, accessed on 16 August 2024), TargetNet (http://targetnet.scbdd.com/home/index/, accessed on 17 August 2024), SuperPred (https://prediction.charite.de/subpages/target_prediction.php, accessed on 17 August 2024), GeneCards (https://www.genecards.org/, accessed on 19 August 2024), OMIM^®^ (https://www.omim.org/, accessed on 19 August 2024), Harmonizome (https://maayanlab.cloud/Harmonizome/, accessed on 19 August 2024), and PharmGKB (https://www.pharmgkb.org/, accessed on 19 August 2024). A minimum confidence score of 0.5 was set for target selection in TargetNet, while other databases had no specific confidence restrictions. To identify and analyze potential neuronal development-related targets, shared targets between four key bioactive compounds (anethole, 2-Propanone, 1-(4-methoxyphenyl)-, silane, ethoxy(dimethylphenyl)-, and para-anisaldehyde diethyl acetal) and neuronal development were identified and visualized using a Venn diagram. Protein-protein interaction (PPI) analysis was conducted using Cytoscape (version 3.9.0), ShinyGO (version 0.76: (http://bioinformatics.sdstate.edu/go76/, accessed on 20 August 2024), and STRING (version 12.0: https://string-db.org/, accessed on 20 August 2024), focusing on shared targets between anethole and neuronal development with a confidence score above 0.4. Cytoscape’s network analyzer tool was used to examine topological features within the network. Gene Ontology (GO) and Kyoto Encyclopedia of Genes and Genomes (KEGG) pathway enrichment analyses were performed using ShinyGO (v0.76), with results visualized in dot plot charts. The analysis was set to “Homo sapiens” as the specified species. KEGG pathway enrichment highlighted the Neurotrophin signaling pathway, suggesting that FVSE compounds may influence genes within this pathway, which are associated with neuronal development and function.

### 4.9. Statistical Analysis

All statistical analyses were conducted using GraphPad Prism version 9.2 for Windows. Data are presented as the mean ± SEM of at least three independent experiments. For post hoc analyses, data were evaluated using either Student’s *t*-test or one-way analysis of variance (ANOVA) followed by Duncan’s multiple comparison test (SPSS version 16.0). Additionally, Student’s *t*-tests with equal sample variance and two-tailed conditions were applied. Statistical significance was set at *p* ≤ 0.05, and differences were considered statistically significant.

## 5. Conclusions

This study demonstrates that *F. vulgare* seed extract (FVSE) promotes neuronal growth, enhances synaptic connections, and improves neuronal survival, all of which are crucial for maintaining cognitive health. The bioactive compounds in FVSE, particularly anethole, positively influence key pathways involved in neuronal survival, neuroprotection, neurodevelopment, and neuroplasticity. Notably, FVSEs effects are mediated through the modulation of neurotrophic factors such as BDNF and NGF, which activate downstream signaling pathways that regulate neuronal growth and resilience. Our findings suggest that FVSE influences the Neurotrophin signaling pathway, particularly through Trk-A and Trk-B receptors, which are critical for neuronal survival and plasticity. FVSEs impact on key signaling components like Bcl2 and GSK3β^ser9^ supports synaptic stability and axonal growth, mechanisms essential for preserving cognitive function and resilience in neurodegenerative diseases (NDs) like Alzheimer’s disease. These insights underscore FVSEs potential as a complementary therapy for managing NDs. Future research, including network pharmacology, computational docking and simulation analyses, and animal studies, is necessary to confirm these findings and assess the clinical applicability of FVSE and its bioactive compounds. The results of these future studies could significantly guide the development of FVSE and its bioactive compounds as potential therapeutic agents for ND treatment.

## Figures and Tables

**Figure 1 ijms-25-12701-f001:**
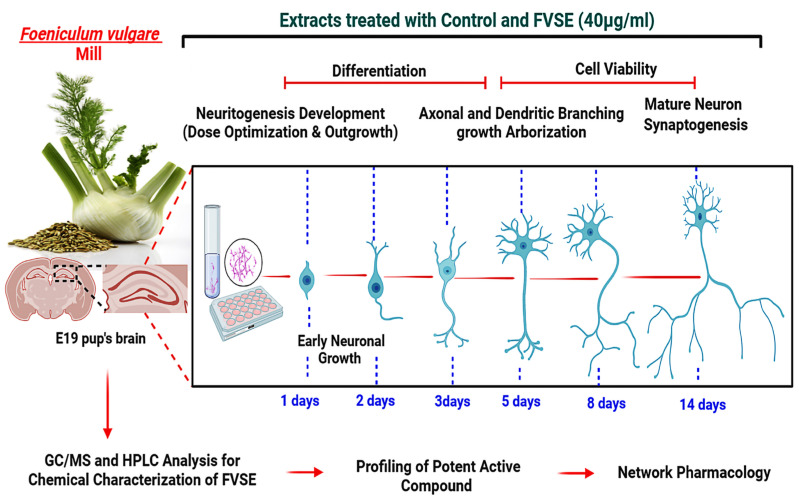
Diagram illustrating the evaluation of FVSE on hippocampal neurons, covering stages from chemical analysis (GC/MS and HPLC) to neuronal differentiation, growth, and synaptogenesis under FVSE (40 µg/mL) treatment.

**Figure 2 ijms-25-12701-f002:**
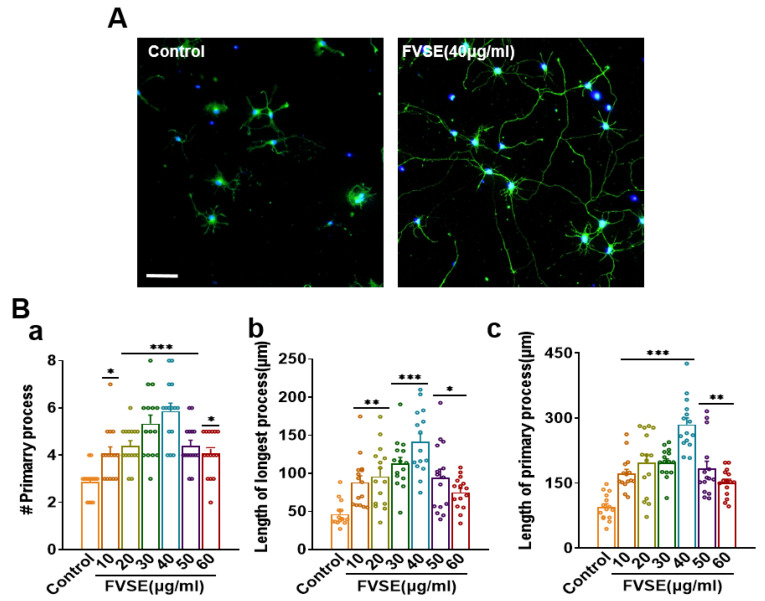
Neurite Outgrowth Quantification in FVSE-treated Neurons: Hippocampal neurons were cultured for three days and treated with FVSE at varying concentrations. Immuno-stained images illustrate neurite outgrowth, with representative images comparing FVSE (40 μg/mL) and vehicle treatments. Morphometric analysis assessed the number of neurites, total neurite length, and the length of the longest neurite. Statistical significance was determined by one-way ANOVA, followed by Tukey’s post hoc test (* *p* < 0.05, ** *p* < 0.01, *** *p* < 0.001). Each experiment was replicated three times (*n* = 3), with 10 neurons per replicate. Data are presented as Mean ± S.E.M.

**Figure 3 ijms-25-12701-f003:**
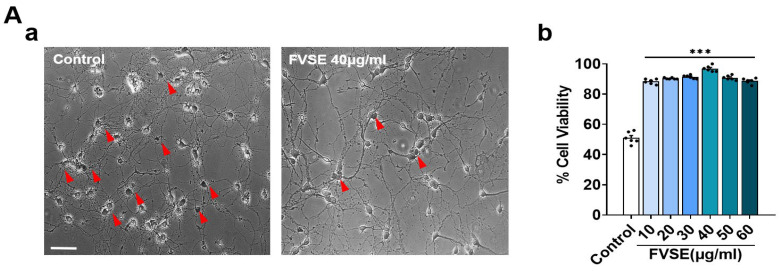
Effects of FVSE on Neuronal Cell Viability. (**A**) Trypan blue-stained images showing viable neurons, with dead neurons indicated by red arrows in panel (**a**). The bar graph (**b**) illustrates the percentage of surviving cells after treatment with varying concentrations of FVSE (10–60 μg/mL). The scale bar represents 50 μm. Neuronal viability was determined by counting the number of unstained (live) cells relative to the total number of assessed cells (living and dead). Results were analyzed using one-way analysis of variance (ANOVA) and are presented as mean ± standard error of the mean (SEM) from three independent studies (*n* = 6, with 400–500 neurons per study). Statistical significance is indicated by *** *p* < 0.001.

**Figure 4 ijms-25-12701-f004:**
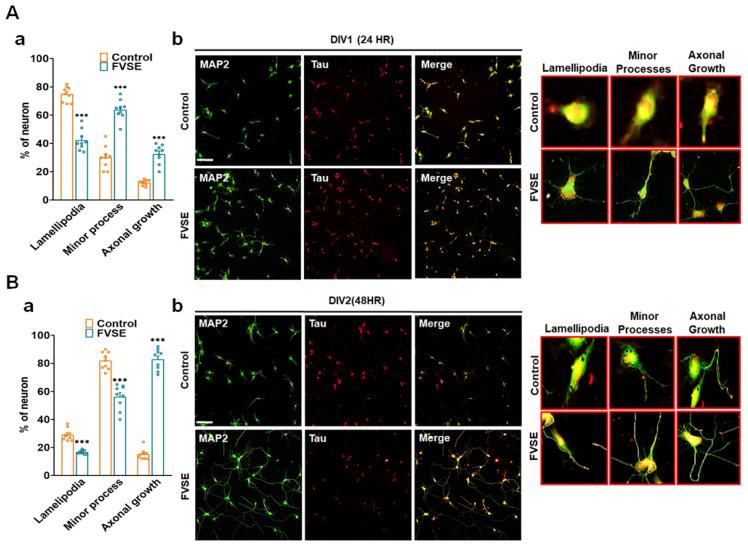
During the initial stages of neuronal differentiation, FVSE was applied at a concentration of 40 μg/mL to assess early developmental maturation. Neurons were fixed at DIV1 and DIV2 and stained with antibodies for MAP2 (green) and tau (red) under consistent culture conditions throughout the experiment. (**A**(-b)) Representative fluorescence image showing early development at 24 h (DIV1), with insets representing normal developmental stages: Stage 1 (lamellipodia formation), Stage 2 (minor process formation), and Stage 3 (neurite outgrowth). (**A**(-a)) Statistical analysis displaying the percentage of neurons reaching each developmental stage at the corresponding time points. (**B**(-b)) Representative fluorescence images taken at 48 h (DIV2), along with the corresponding statistical analysis (**B**(-a)). Scale bars represent 50 μm and 10 μm (insets). Values in the bar graphs are presented as mean ± SEM from 300 to 400 neurons. Statistical significance compared to the vehicle is indicated by *** *p* < 0.001.

**Figure 5 ijms-25-12701-f005:**
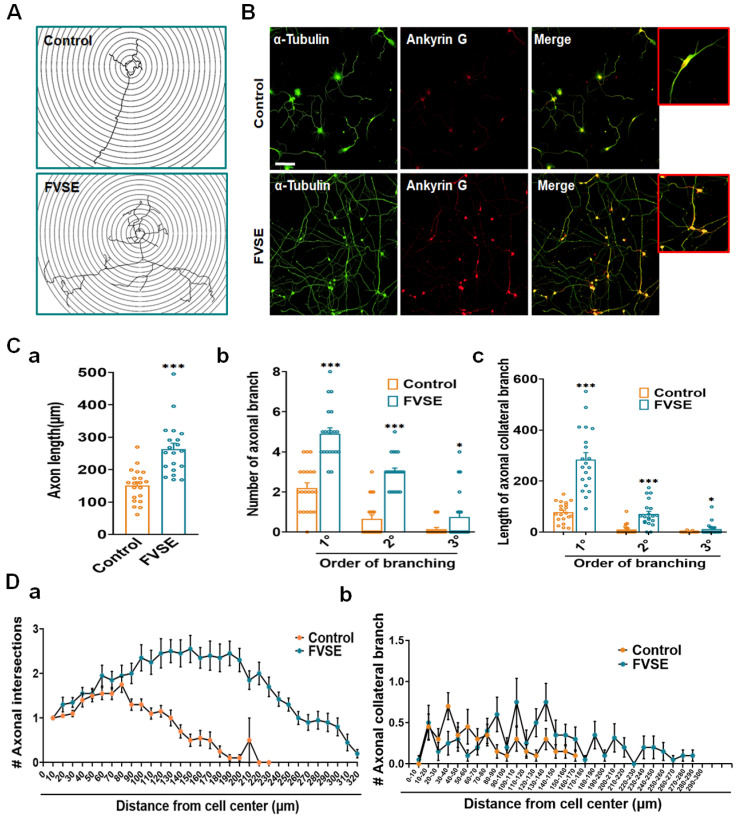
FVSE Enhances Axon Morphogenesis in Hippocampal Neurons. Hippocampal neurons were cultured for 5 days under the conditions described in Figure 2. After culture, cells were fixed and subjected to double immunostaining with ankyrin G (red) and α-tubulin (green). Ankyrin G, located at the axon initial segment (AIS), was used to identify axons, while α-tubulin marked dendrites. Representative fluorescence images of DIV5 cultures were used to assess neuronal morphology, with a scale bar of 50 μm. Morphometric analysis includes (**A**) sholl analysis and (**B**) immune-fluorescent images comparing FVSE-treated and control neurons. Quantitative measurements showed significant increases in FVSE-treated neurons for (**C**(-a)) axon length, (**C**(-b)) number of axonal branches by branching order, and (**C**(-c)) axonal collateral branch length by branching order. Sholl analysis also highlighted enhanced axonal complexity in FVSE-treated neurons, as observed in (**D**(-a)) the number of axonal intersections and (**D**(-b)) the total length of axonal branches. Data are presented as mean ± SEM (from three independent biological replicates). Statistical significance compared to the control is indicated by * *p* < 0.05 and *** *p* < 0.001 (Student’s *t*-test).

**Figure 6 ijms-25-12701-f006:**
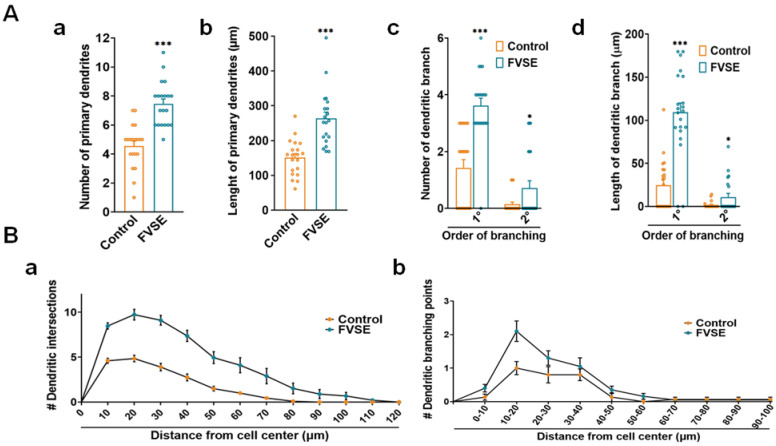
FVSE Promotes Dendritic Morphogenesis in Neurons. Representative photomicrographs for dendritic morphogenesis, similar to those in Figure 5B, show the differences between FVSE-treated and control neurons. Morphometric analysis includes (**A**(-a)) the number of primary dendrites, (**A**(-b)) the total length of primary dendrites, (**A**(-c)) the number of dendritic branches, and (**A**(-d)) the total length of dendritic branches. Beyond that, Sholl analysis illustrates (**B**(-a)) the number of dendritic intersections and (**B**(-b)) the branching points at various distances from the cell center. Bar graphs present mean values ± S.E.M. (*n* = 3, 10 neurons per group). Statistical significance relative to the vehicle control is denoted by * *p* < 0.05 and *** *p* < 0.001.

**Figure 7 ijms-25-12701-f007:**
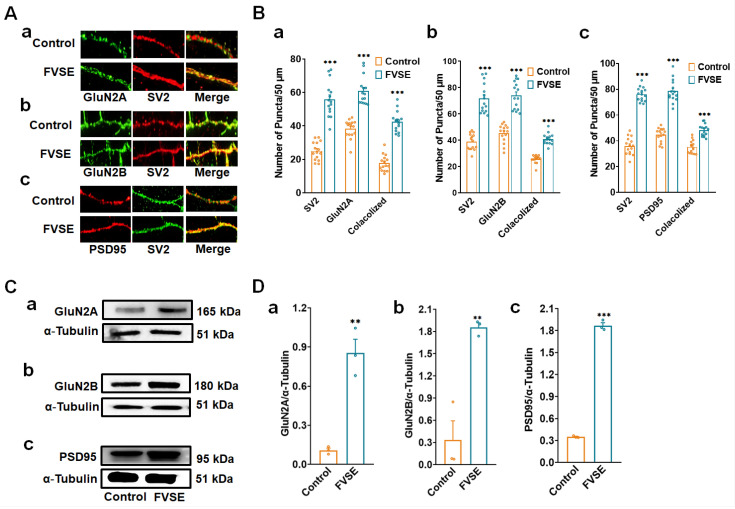
Impact of FVSE on Synaptic Connections. (**A**) Fluorescence microscopy images illustrating synaptic connections with dual labeling of SV2 (green) and NMDA receptor subunits (GluN2A and GluN2B) or PSD95 (red), showing co-localization (yellow puncta) in control and FVSE-treated neurons. Panels a–c show the synaptic co-localization of SV2 with GluN2A, GluN2B, and PSD95, respectively, highlighting an increase in synapse density with FVSE treatment. The scale bar represents 2 µm. (**B**) Quantification of puncta density for SV2 and each respective marker in a 50 µm segment. Panels (**B**(-a))–(**B**(-c)) display the density of SV2 co-localized with GluN2A, GluN2B, and PSD95, respectively, showing significant increases in co-localized puncta in FVSE-treated neurons compared to controls. (**C**) Immunoblotting analysis of neuronal lysates from DIV14, verifying the expression levels of GluN2A, GluN2B, and PSD95 in FVSE-treated versus control neurons, with α-tubulin as a loading control. Panels (**C**(-a))–(**C**(-c)) present the immunoblotting results for GluN2A, GluN2B, and PSD95, respectively, with increased expression in FVSE-treated neurons. (**D**) Bar graphs quantifying the relative intensity of GluN2A, GluN2B, and PSD95 normalized to α-tubulin, as shown in panels (**D**(-a))–(**D**(-c)). These results demonstrate a significant upregulation of synaptic components in FVSE-treated neurons, supporting enhanced synaptic connectivity and plasticity. Statistical data are presented as mean ± standard error of the mean (S.E.M.) from three independent experiments (*n* = 3, with 10 neurons per experiment). Student’s *t*-tests were used to determine statistical significance (** *p* < 0.01 and *** *p* < 0.001).

**Figure 8 ijms-25-12701-f008:**
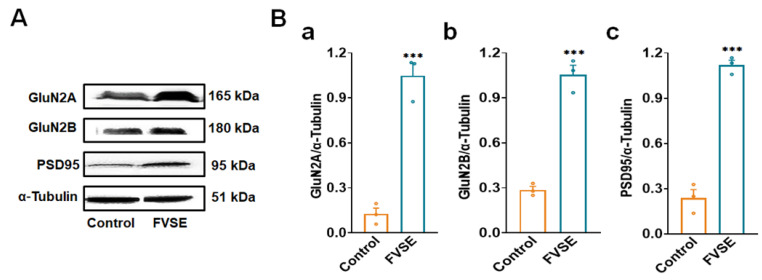
Analysis of Synaptic Protein Expression in the Cortex of 5-Week-Old ICR Mice Treated with FVSE. (**A**) Representative Western blot images showing the levels of GluN2A, GluN2B, PSD95, and the loading control α-Tubulin in cortical tissue from control and FVSE-treated groups. (**B**) Quantitative analysis of Western blot band intensity normalized to α-Tubulin for (a) GluN2A, (b) GluN2B, and (c) PSD95. Data are expressed as mean ± S.E.M. from four mice per group. Statistically significant differences compared to the control group are indicated by *** *p* < 0.001 (Student’s *t*-test).

**Figure 9 ijms-25-12701-f009:**
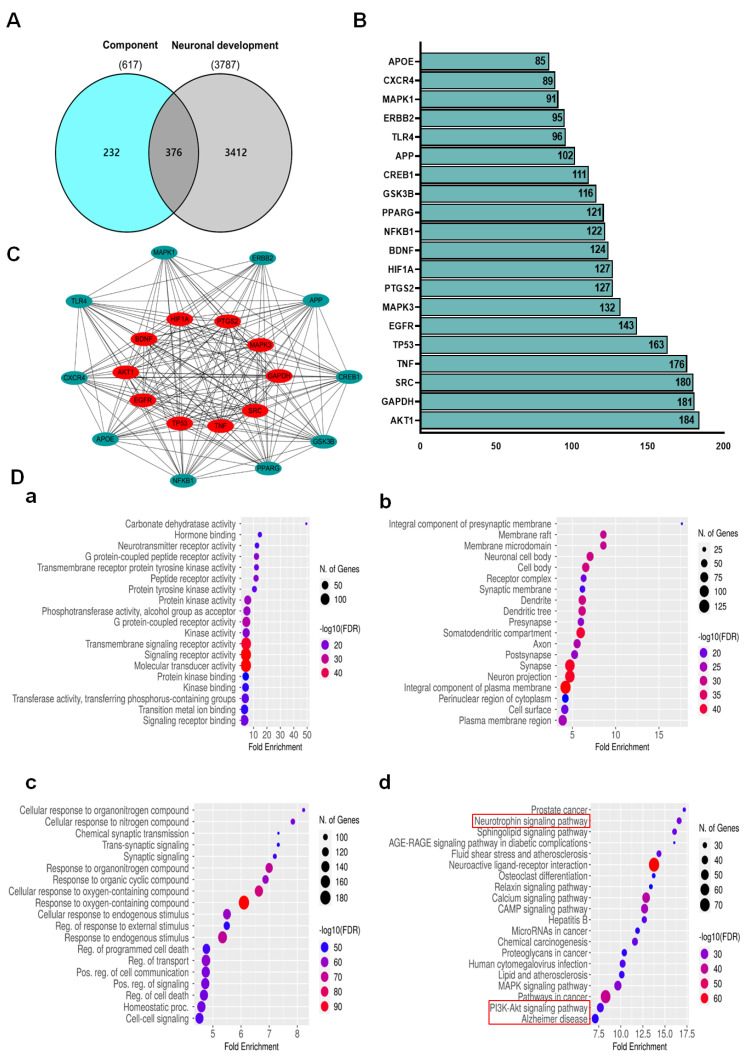
Network and Enrichment Analysis of FVSE Bioactive Compounds and Neuronal Development Targets. (**A**) Venn diagram showing the overlap between genes associated with FVSE bioactive compounds and those involved in neuronal development, identifying 376 common genes. (**B**) Protein-protein interaction (PPI) network of the 20 common genes, with key hub genes highlighted in red, including AKT1, SRC, and TP53. (**C**) Bar chart displaying the top 20 genes related to neuronal development, with critical genes labeled. (**D**) Dot plots representing enrichment analysis of common target genes, showing significant involvement in (a) biological processes, (b) cellular components, (c) molecular functions, and (d) KEGG pathways, highlighting pathways such as Neurotrophin signaling, PI3K-AKT signaling, and Alzheimer’s disease pathways.

**Figure 10 ijms-25-12701-f010:**
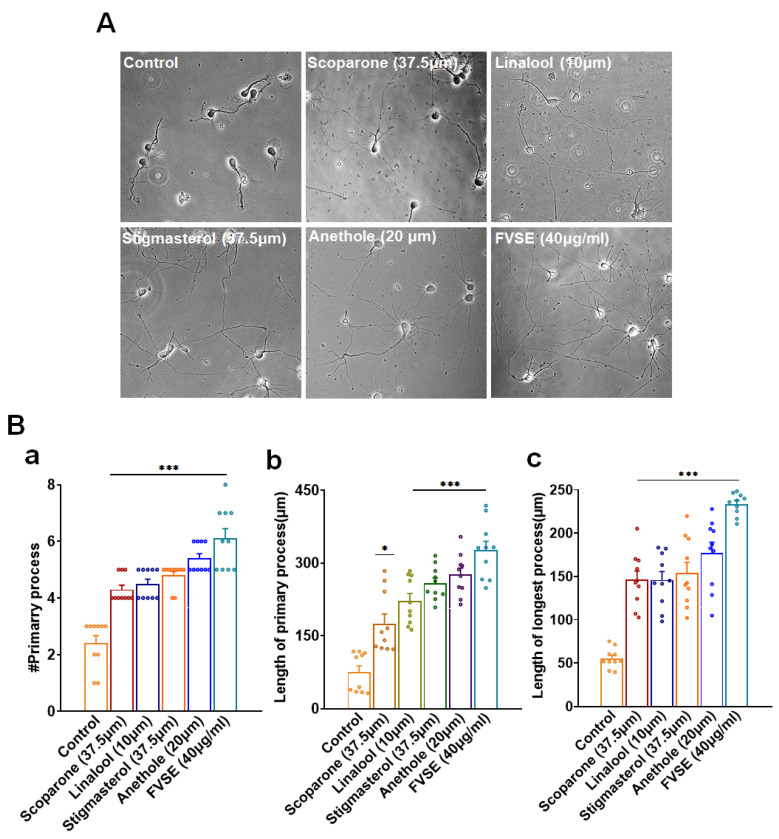
(**A**) Bright-field images illustrating the neuritogenetic effects of various treatments, including Scoparone (37.5 μM), Linalool (10 μM), Stigmasterol (37.5 μM), Anethole (20 μM), and FVSE (40 μg/mL), compared to the control. Neurite outgrowth is visible across the different treatments, with the scale bar representing 50 μm. (**B**) Quantitative analysis of morphometric parameters, including (a) the number of primary neurites per neuron, (b) the length of the primary neurites, and (c) the length of the longest neurite. Data are presented as the mean ± standard error of the mean (S.E.M.) from three independent experiments (*n* = 3, with 10 neurons per experiment). Statistical significance was determined using one-way analysis of variance (ANOVA), with * *p* < 0.05 and *** *p* < 0.001 indicating levels of significance.

**Figure 11 ijms-25-12701-f011:**
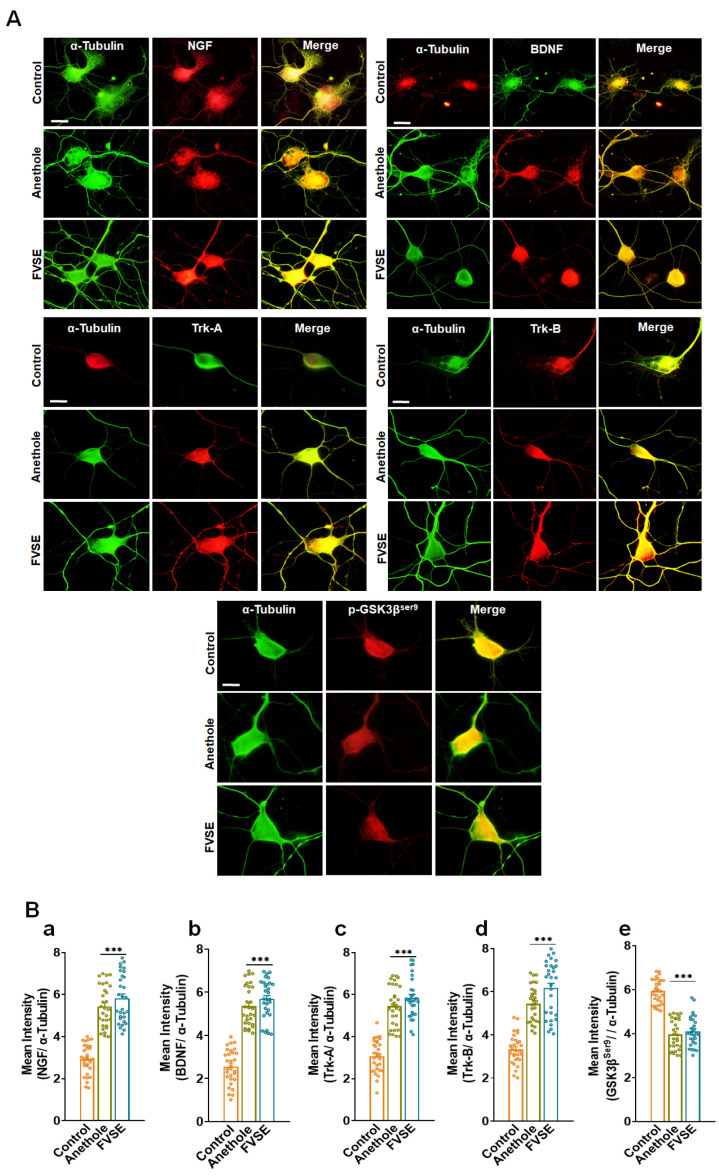
Immunofluorescence Characterization of Neurotrophin Signaling Proteins in Primary Cultured Neurons. (**A**) Representative fluorescence images of neurons labeled with α-tubulin (green) as a structural marker, alongside neurotrophin proteins (red): NGF, BDNF, Trk-A, Trk-B, and GSK3βser9. Merged images show areas of co-localization (yellow), indicating overlap between α-tubulin and the specific neurotrophin markers. Scale bar represents 50 µm. (**B**) Quantitative analysis of mean fluorescence intensity for neurotrophin markers relative to α-tubulin expression: (a) NGF, (b) BDNF, (c) Trk-A, (d) Trk-B, and (e) GSK3βser9. Data are expressed as mean ± S.E.M. from three independent experiments with 30 neurons per group. Statistical significance is denoted by *** *p* < 0.001 (Student’s *t*-test).

**Figure 12 ijms-25-12701-f012:**
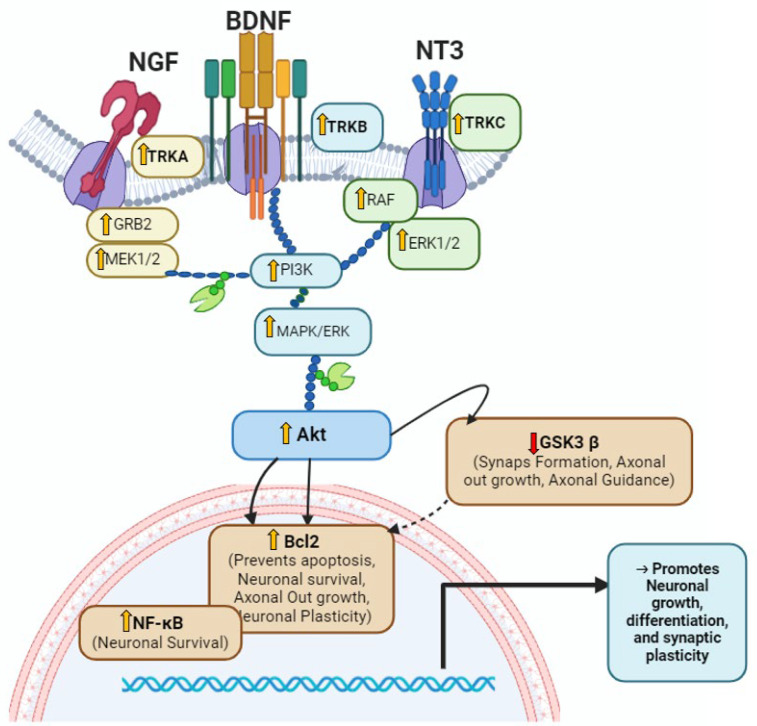
The diagram illustrates the neurotrophic activity of FVSE via activation of the Trk-B receptor in the Neurotrophin signaling pathway. This activation triggers the PI3K/Akt pathway, which regulates Bcl2, NF-κB, and GSK3β^ser9^ to promote neuronal survival, differentiation, and synaptic plasticity. Solid arrows represent activation, and blunt arrows indicate inhibition.

## Data Availability

The data that support the findings of this study are available from the corresponding author upon reasonable request.

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
