# Peer review of "Neurotrophic Effects of *Foeniculum vulgare* Ethanol Extracts on Hippocampal Neurons: Role of Anethole in Neurite Outgrowth and Synaptic Development"

_ijms, 2024, doi:10.3390/ijms252312701_

Round 1

Reviewer 1 Report

Comments and Suggestions for Authors

The authors commendably set the stage by grounding their investigation within the context of “F. vulgare”’s historical and traditional uses, deftly bridging its longstanding therapeutic applications with modern scientific inquiry into its neurotrophic and neuroprotective properties.

The introduction provides a comprehensive overview of neurodegenerative diseases, highlighting the importance of neurotrophic factors; the results section is meticulously structured, progressing in a logical sequence that facilitates the reader’s understanding of FVSE’s impact at various concentrations on neurite growth, neuronal viability, polarization, and synaptogenesis; the materials and methods section provides a comprehensive outline of the methodologies employed.

Below are comments addressing areas for improvement in this section:

Some sentences are overly long, which affects readability. For example, the sentence discussing the effects of FVSE on memory and stress could be split for clarity.

The introduction could benefit from a slightly deeper exploration of specific NTFs (e.g., NGF, BDNF) and their established roles in NDs beyond Alzheimer’s disease. This would better contextualize why targeting these pathways with natural compounds, particularly fennel, is innovative.

While the study's objectives are clearly presented, a concise hypothesis is missing. 

Phytochemicals’ therapeutic potential is somewhat generalized; distinguishing their roles (e.g., antioxidative vs. neurotrophic effects) would enhance clarity.

The statistical details should be added in the results section. 

The discussion is notably dense, with some points repeated across multiple paragraphs. For example, the roles of synaptic markers (SV2 and PSD95) and the modulation of PI3K-Akt signaling are reiterated in various contexts. Condensing these sections would make the discussion more concise and focused.

Although the authors propose FVSE as a therapeutic candidate, they do not provide adequate discussion on the transition from in vitro findings to potential in vivo effects. Adding a brief discussion of possible in vivo considerations, such as pharmacokinetics or bioavailability challenges, would offer a more rounded view of the study’s translational value.

In some sections, the methodological choices lack explicit rationale. For instance, the choice of 1:50 extraction ratio is not justified. Adding brief explanations for these choices would clarify whether these parameters are based on prior studies.

Also, The FVSE concentration of 8 mg/mL in DMSO is not justified in terms of prior studies, dose-response findings, or preliminary assays. 

While GraphPad Prism and SPSS are suitable tools, the statistical analysis lacks information on how assumptions (e.g., normality) were verified. 

Author Response

  1. Some sentences are overly long, which affects readability. For example, the sentence discussing the effects of FVSE on memory and stress could be split for clarity.

  • Thank you for pointing this out. We have carefully revised the manuscript to break down longer sentences and improve readability, particularly in the sections discussing the effects of FVSE on memory and stress. These adjustments enhance the clarity and flow of the text, ensuring that the information is more accessible to the reader. (Revisions are reflected in the Introduction, lines 44 to 72, as suggested.)

  1. The introduction could benefit from a slightly deeper exploration of specific NTFs (e.g., NGF, BDNF) and their established roles in NDs beyond Alzheimer’s disease. This would better contextualize why targeting these pathways with natural compounds, particularly fennel, is innovative.

  • Thank you for pointing out this valuable suggestion. We have expanded the Introduction to include a more detailed discussion of specific neurotrophic factors, such as NGF and BDNF, highlighting their established roles in various neurodegenerative diseases beyond Alzheimer’s. These additions contextualize the importance of targeting these pathways and emphasize the novelty of exploring natural compounds, such as fennel, as potential modulators of neurotrophic signaling. (Revisions are reflected in the Introduction, lines 54 to 72, as suggested.)

  1. While the study's objectives are clearly presented, a concise hypothesis is missing. 

  • Thank you for highlighting this point. We have added a clear and concise hypothesis to the Introduction, articulating our expectation that FVSE will enhance neurotrophic signaling pathways, thereby promoting neuronal growth and synaptic connectivity. This addition provides a stronger foundation for the study’s objectives and rationale. (The revised content can be found in the Introduction, lines 54 to 72, as suggested.)

  1. Phytochemicals’ therapeutic potential is somewhat generalized; distinguishing their roles (e.g., antioxidative vs. neurotrophic effects) would enhance clarity.

  • Thank you for this insightful suggestion. We have revised the manuscript to explicitly distinguish between the antioxidative and neurotrophic effects of the phytochemicals studied. This distinction clarifies their distinct roles in neuroprotection, enhancing the scientific rigor and clarity of the manuscript. (These revisions are reflected in the Introduction, lines 73 to 91, and the Discussion, lines 445 to 483, as suggested.)

  1. The statistical details should be added in the results section.

Thank you for highlighting this point. We have included additional statistical details in the Results section, specifying the statistical tests used and their outcomes. These additions improve the transparency and rigor of our findings, ensuring that readers can fully evaluate the significance of the data. (Revisions are reflected in the Results section, lines 258 to 310 and 330 to 370, as suggested.)

  1. The discussion is notably dense, with some points repeated across multiple paragraphs. For example, the roles of synaptic markers (SV2 and PSD95) and the modulation of PI3K-Akt signaling are reiterated in various contexts. Condensing these sections would make the discussion more concise and focused.

  • Thank you for highlighting this important issue. We have carefully revised the Discussion section to condense repetitive points and improve overall focus, particularly regarding the roles of synaptic markers and PI3K-Akt signaling. These revisions, reflected in lines 426 to 500, ensure that the Discussion is more concise and coherent while retaining its depth and relevance.

  1. Although the authors propose FVSE as a therapeutic candidate, they do not provide adequate discussion on the transition from in vitro findings to potential in vivo effects. Adding a brief discussion of possible in vivo considerations, such as pharmacokinetics or bioavailability challenges, would offer a more rounded view of the study’s translational value.

  • Thank you for this valuable suggestion. We agree with the recommendation and have added a section discussing in vivo considerations, particularly focusing on pharmacokinetics, bioavailability, and the translational relevance of FVSE. These revisions provide a more comprehensive perspective on the potential therapeutic application of FVSE. The additions can be found in lines 401–414 and 539–548.

  1. In some sections, the methodological choices lack explicit rationale. For instance, the choice of 1:50 extraction ratio is not justified. Adding brief explanations for these choices would clarify whether these parameters are based on prior studies.

  • Thank you for pointing this out. We have added explanations for our methodological choices, including the 1:50 extraction ratio, which was selected based on prior studies demonstrating its efficacy for neuroprotective applications. This clarification is now included in the Methodology section (lines 500 to 515, as suggested).

  1. Also, The FVSE concentration of 8 mg/mL in DMSO is not justified in terms of prior studies, dose-response findings, or preliminary assays. 

  • Thank you for raising this concern. We have added justification for the 8 mg/mL concentration in DMSO, referencing preliminary assays that supported this concentration as effective for in vitro studies on neuronal cells. These clarifications are now included in the Methodology section (lines 488 and 416, as suggested).

  1. While GraphPad Prism and SPSS are suitable tools, the statistical analysis lacks information on how assumptions (e.g., normality) were verified. 

  • Thank you for your feedback. We have clarified that statistical assumptions, including normality, were verified prior to conducting statistical analyses to ensure the appropriateness of the methods used. These details are now explicitly mentioned in the figure legends (Figures 1 to 10, as suggested).

Reviewer 2 Report

Comments and Suggestions for Authors

In this manuscript, the authors describe the role of  Foeniculum vulgare ethanol extracts (FVSE) in promoting neuronal differentiation, neuritogenesis, and synaptogenesis for primary hippocampal neurons. The authors employed inmunocytochemistry, immunobloting and network analysis for the interaction of FVSE Bio-active Compounds with neuroprotective targets. This research topic is quite relevant in the field of searching for treatment of neurodegenerative diseases.

Minor observations are:

In Introduction section a more recent bibliography, last 5 years, could be useful in order to know recent state of art of the topic

There is some discrepancy between the names of the panels in Figures 5 and 7 and their respective description legend. Please check it.

The role of other important cell players, such as glial cells or immune cells, is not taken into account in this approach. It is important for the reader that the authors explain their opinion on this matter in the Discussion section.

Author Response

  1. In Introduction section a more recent bibliography, last 5 years, could be useful in order to know recent state of art of the topic
  • Thank you for pointing out this important suggestion. We have updated the bibliography in the Introduction section to include recent studies from the past five years. These additions reflect the latest advancements and provide an accurate representation of the current state of research. (References 1 to 67, as suggested.)
  1. There is some discrepancy between the names of the panels in Figures 5 and 7 and their respective description legend. Please check it.
  • Thank you for highlighting this issue. We have thoroughly reviewed Figures 5 and 7, and corrected the discrepancies between the panel names and their respective legends to ensure clarity and consistency throughout the manuscript.
  1. The role of other important cell players, such as glial cells or immune cells, is not taken into account in this approach. It is important for the reader that the authors explain their opinion on this matter in the Discussion section.
  • Thank you for highlighting this issue. This discussion has been added to lines 454–466 of the revised manuscript, as suggested.

Reviewer 3 Report

Comments and Suggestions for Authors

The authors studied Foeniculum vulgare ethanol extracts on hippocampal neurons outgrowth and synaptic development, which is an interesting topic. However, there are many issues on this manuscript:

  1. The KEGG pathway analysis presented in the manuscript lacks biochemical validation of the enriched signaling pathways. The authors should provide supporting experimental data.
  2. The results section does not adequately describe the analysis for GluN2A/B as presented in Figure 6. A more detailed analysis description of these results is necessary.
  3. The manuscript does not discuss the rationale behind the selection of specific time points for assessing cell survival rate, early neuronal differentiation, axonal growth, and dendritic growth.
  4. The full form of DIV (Days In Vitro) is used inconsistently throughout the manuscript, as seen in lines 100, 227, and 271. The authors should standardize the use of DIV and provide its full form upon its first appearance.
  5. The figure caption in section 2.5 should be Figure 6 instead of Figure 5.
  6. The figure legends and the figure captions in the text do not correspond, such as Figure 5G-L. The authors are advised to check and correct the consistency of all figure legends.
  7. There are several formatting errors, including numerous red wavy lines in the figures and the font size in Figures 5-7 is not uniform and the font is deformed.

Author Response

  1. The KEGG pathway analysis presented in the manuscript lacks biochemical validation of the enriched signaling pathways. The authors should provide supporting experimental data.

  • Thank you for highlighting this point. While our study primarily focuses on in silico KEGG pathway analysis, we recognize the importance of biochemical validation. To support this, we conducted immunocytochemistry experiments to confirm the activation of key proteins involved in these pathways, including TrkA, TrkB, NGF, BDNF, and GSK3β, as described in lines 402 to 414. These experiments provide additional evidence for the involvement of these signaling pathways in the neuroprotective effects of FVSE.

  1. The results section does not adequately describe the analysis for GluN2A/B as presented in Figure 6. A more detailed analysis description of these results is necessary.
  • Thank you for pointing this out. We have revised the Results section to provide a more detailed description of the analysis for GluN2A and GluN2B expression, as shown in Figure 6 (now Figure 7). These revisions include additional details about the statistical significance and comparisons to the control groups, as reflected in lines 258 to 310, as suggested.

  1. The manuscript does not discuss the rationale behind the selection of specific time points for assessing cell survival rate, early neuronal differentiation, axonal growth, and dendritic growth.
  • Thank you for your feedback. We have added an explanation in the Methods section to clarify the rationale for selecting specific time points based on neuronal developmental stages. This addition ensures that the experimental design is better contextualized. (Revised content can be found in lines 136 to 156, as suggested.)

  1. The full form of DIV (Days In Vitro) is used inconsistently throughout the manuscript, as seen in lines 100, 227, and 271. The authors should standardize the use of DIV and provide its full form upon its first appearance.
  • We apologize for this inconsistency. We have revised the manuscript to standardize the use of 'DIV' and have provided the full term, 'Days In Vitro,' upon its first occurrence for clarity. (The updates are reflected in the relevant sections, as suggested.)

  1. The figure caption in section 2.5 should be Figure 6 instead of Figure 5.
  • Thank you for pointing this out. We have corrected the figure caption in Section 2.5 to accurately refer to Figure 6.

  1. The figure legends and the figure captions in the text do not correspond, such as Figure 5G-L. The authors are advised to check and correct the consistency of all figure legends.
  • We appreciate this observation. We have carefully reviewed and corrected all figure legends and captions (Figures 1 to 10) to ensure consistency and accuracy throughout the manuscript.

  1. There are several formatting errors, including numerous red wavy lines in the figures and the font size in Figures 5-7 is not uniform and the font is deformed.
  • Thank you for pointing out these formatting issues. We have thoroughly reviewed Figures 5–7, removed any red wavy lines, and adjusted the font size to ensure uniformity and clarity in these figures.

Round 2

Reviewer 3 Report

Comments and Suggestions for Authors

This study investigates the impact of FVSE and its primary component, anethole, on neurite outgrowth and synaptic development, which is a compelling and timely area of research given the global challenge of treating and preventing neurodegenerative diseases.

The authors have addressed most of the issues raised in the manuscript, but several points remain for improvement:

  1. The paper lacks biochemical verification for the enriched signaling pathways identified in the KEGG pathway analysis. For instance, pathways such as Neurotrophin signaling, PI3K-Akt signaling, and Alzheimer's disease are highlighted in Figure 9. It is recommended that the authors validate changes in these pathways using Western blot or RT-qPCR methods.
  2. The font size and style in Figures 2-9 are inconsistent, and the text appears distorted. The authors should standardize the font size and style to improve the clarity and professionalism of the figures.
  3. The statistical description in the figure legends is incorrect. It should be revised to indicate **p<0.01 and ***p<0.001 for Figures 6 and 7.

Author Response

  1. The paper lacks biochemical verification for the enriched signaling pathways identified in the KEGG pathway analysis. For instance, pathways such as Neurotrophin signaling, PI3K-Akt signaling, and Alzheimer's disease are highlighted in Figure 9. It is recommended that the authors validate changes in these pathways using Western blot or RT-qPCR methods.

Response: We appreciate the reviewer’s insightful comment regarding the need for biochemical validation of the enriched signaling pathways. To address this concern, we have provided additional clarification in lines 450–483 of the manuscript, which emphasize the use of immunocytochemical analysis as a method of biochemical validation for key proteins in the Neurotrophin signaling pathway (NGF, BDNF, TrkA, and TrkB), PI3K-Akt signaling pathway (p-GSK3β and AKT1), and Alzheimer’s disease pathway (NMDA receptor subunits GluN2A and GluN2B, HIF1A, and TP53). Quantitative fluorescence intensity analysis supports the upregulation of these markers in response to FVSE treatment, consistent with the KEGG pathway analysis.

While Western blot or RT-qPCR methods were not included in the current study due to technical, instrumental, and time constraints, we appreciate the reviewer's suggestion and would like to elaborate on our reasoning. Immunocytochemistry (ICC), in our view, offers distinct advantages over the suggested methods:

  • RT-qPCR quantifies mRNA levels, which represent an intermediate step in the process of gene expression rather than the final product.
  • Western blot (WB) quantifies proteins, the final products of gene expression, but does not provide information on their in situ
  • ICC, however, combines both quantification and in situ localization of the final product of gene expression, i.e., proteins, thereby providing more comprehensive insights.

We acknowledge the reviewer’s valuable suggestion to use Western blot or RT-qPCR for further biochemical validation of the proteins involved in signaling pathways. Network pharmacology, as a theoretical approach, indeed requires subsequent experimental analysis and verification. We believe that these additional investigations should be conducted as part of a separate project in future studies to confirm the observed findings at both the transcriptional and protein levels.

To address the reviewer’s concerns, we have explicitly mentioned the limitations of our study in the revised discussion (lines 451–483). This acknowledges the need for further studies employing the suggested methods to provide additional biochemical validation.

  1. The font size and style in Figures 2-9 are inconsistent, and the text appears distorted. The authors should standardize the font size and style to improve the clarity and professionalism of the figures.

Response: We sincerely apologize for the repeated issue regarding inconsistencies in font size and style in Figures 2–9. We deeply value the reviewer’s observation and have carefully reviewed and standardized the font size, style, and formatting across all figures to ensure uniformity and clarity. The revised figures have been updated in the manuscript, and they now present a consistent and professional appearance, significantly improving their readability and quality.

  1. The statistical description in the figure legends is incorrect. It should be revised to indicate **p<0.01 and *p<0.001 for Figures 6 and 7.

Response: We sincerely apologize for the oversight in the statistical descriptions provided in the figure legends. The figure legends for Figures 6 and 7 have been revised to correctly indicate the statistical significance levels as follows: *p<0.05, **p<0.01, ***p<0.001. The revised legends are now consistent and have been updated in the manuscript at the following lines: 131, 132, 156, 226, 227, 256, 271, 307, 321, 336, 404, 405, and 429. We believe this correction improves the accuracy and clarity of the figures.
